# Effect of Harvest Time on the Seed Yield and Quality of *Kengyilia melanthera*

Yao Ling [1,†], Shuai Yuan [1,2,†], Yanli Xiong [1], Shuming Chen [1], Junjie Feng [1], Junming Zhao [1], Chenglin Zhang [1], Xiong Lei [2], Minghong You [2], Shiqie Bai [2] and Xiao Ma [1,*]

1   College of Grassland Science and Technology, Sichuan Agricultural University, Chengdu 611130, China
2   Sichuan Academy of Grassland Sciences, Chengdu 611743, China
*   Correspondence: maroar@126.com
†   These authors contributed equally to this work.

**Abstract:** *Kengyilia melanthera* is one of the most important forages, and has received significant attention as a desirable ecological pioneer grass for conserving grasslands and mitigating degradation pressure in a region. It is widely distributed in the alpine sandy meadow zone of the Eastern Qinghai–Tibet Plateau (QTP). Therefore, determining the optimal harvest time of this species is critical. A two-year field experiment (2016–2017) was utilized in this study to evaluate the impact of eight harvest times on the seed yield and quality of *K. melanthera* 'Aba'. The results show that the fresh weight (FW), dry weight (DW), seed yield (SY), thousand-grain weight (TGW), accelerated aging germination percentage (AAGP), and dehydrogenase activity (DA) of seeds increased with the extension of harvest time in two trial years, while the water content (WC) of seeds decreased. In both trial years, the germination percentage (GEP), germination potential (GP), and acid phosphatase activity (APA) of seeds increased initially and then decreased with the extension of harvest time. The comprehensive evaluation of the membership function and hierarchical clustering revealed that the 30th day after peak anthesis was the optimal harvest time for both 2016 and 2017. During this time, FW, DW, SY, TGW, AAGP, DA, WC, GEP, GP, and APA reached their optimal value, with 0.815 g, 0.518 g, 1773.67 kg·hm$^{-2}$, 143.81%, 66%, 16.23 μg·mL$^{-1}$, 36.01%, 87%, 85%, and 2.50 nmol·min$^{-1}$ in 2016 and 0.805 g, 0.515 g, 1847.42 kg·hm$^{-2}$, 133.96%, 65%, 16.52 μg·mL$^{-1}$, 36.04%, 86.5%, 83.5%, and 2.55 nmol·min$^{-1}$ in 2017, respectively. This study uncovers several high-efficiency and long-term technological solutions for cultivating high yields and quality production of *K. melanthera* in East QTP.

**Keywords:** harvest time; *Kengyilia melanthera*; Qinghai–Tibet Plateau; seed quality; seed yield

## 1. Introduction

The Qinghai-Tibetan Plateau (QTP) has been referred to as the "roof of the world" and the "third pole", playing an important role in the terrestrial ecosystem and providing crucial ecological services [1]. However, desertification of the alpine grassland ecosystem in the Eastern QTP has been advancing in recent decades, owing to both human and natural influences [2,3]. Since the early 2000s, the Chinese government has implemented a flurry of policies and initiatives aimed at protecting the ecological environment, reducing degraded grassland regions, and strengthening the biodiversity conservation process [4]. Among them, the adoption of phytorestoration technology is often regarded as one of the simplest and least expensive measures of recovering degraded meadows to a somewhat natural state [5,6]. However, due to a scarcity of suitable grass species and abundantly superior seeds in the QTP, the implementation of this method is severely limited. Notably, *Kengyilia melanthera*, a perennial grass of the Triticeae tribe, is commonly cultivated in the alpine sandy meadows of the Western Sichuan Plateau, and its optimum habitat conditions are at an altitude of 3000–5000 m [7]. Its adaptation to desert environments makes it unique. It has a reasonably high fodder value, and is thus regarded as a key grass species

in the Northwestern Sichuan Plateau for sand fixation, restoration of grassland vegetation, conservation of the ecological environment, and desertification control [8].

Due to its high stress tolerance, *K. melanthera* has recently been used to reverse desertification in parts of the QTP. However, a great number of technical issues were discovered throughout the process of *K. melanthera* seed production, which frequently result in seed yield and quality not fulfilling the specified criteria [9]. As a result, this shortcoming restricts its practical development. The yield of the seed is determined by genetic material as well as environmental factors [10,11]. The optimal harvest time is regarded as a measure of extrinsic factors that can effectively and consistently promote seed yield and quality [12]. According to previous studies, discrepancies in grass seed maturity and shattering might result in harvested seed yields that are only 10 to 20% of their full yield [13]. In addition, seed shattering is considered a physiological phenomenon that occurs towards the mature stage, which can promote improved reproduction [14]. On the other hand, decreasing this unfavorable process is a crucial approach to assure increased profitability during seed production [15,16]. Therefore, determining an optimal harvest time is critical to reaching the level anticipated by the sustainable development plan. At present, only a few research projects have been targeted to studying the effect of harvest time on seed yield and quality in sympatric grass species, such as *K. melanthera*, in the QTP. According to Xie et al.'s [17] study on wild *Elymus nutans* in Tibet, the ideal harvest timing (28th day after peak anthesis) may not only maximize seed yield and fresh and dry weight, but also reduce water content to a comparatively low level. Furthermore, Mao et al.'s [18] study predicted the influence of harvest time on *E. sibiricus*. They found that from 7 to 35 days after peak anthesis, the percentage of seed standard germination and dehydrogenase activity increased at first, then fell dramatically, reaching a peak value on the 27th day following peak anthesis.

Breeding of *K. melanthera* have traditionally been accomplished by multi-year quality selection or recurrent selection in germplasm from wild accessions. This results in a rather basic genetic architecture; hence, the short-term genetic improvement of seed dropping is difficult to achieve. Therefore, a suitable harvest time can not only achieve optimal maturity or remarkable qualities, but also be a vital measure to assess seed production balance. The common indicators for measuring seed quality include germination percentage, germination potential, accelerated aging germination percentage, and seed enzyme activity such as dehydrogenase and acid phosphoesterase activity. Additionally, the water content of seeds and thousand-grain weight are usually used to judge the yield and quality of seeds. These indices therefore were considered critical for determining the optimal harvest time [19].

The requirement for long-term management of degraded grassland on the QTP has driven up demand for high-yielding and high-quality *K. melanthera* seeds. However, the current research on the yield and quality of *K. melanthera* has been mostly concerned with nitrogen fertilizers [20] and seeding time [21]. There was, therefore, no relevant data on seed management based on harvest time prior to the present study. To overcome these obstacles, it was necessary to conduct a two-year investigation program on the influence of different harvest times on seed quality and yield in the Northwest Sichuan plateau. This will help to determine an optimal harvest time, and thus provide a more effective strategy for increasing seed quality and yield of *K. melanthera* 'Aba'.

## 2. Materials and Methods

### 2.1. Experimental Sites

A two-year (from 2016 to 2017) field plot experiment was conducted at Hongyuan County's second farm, located in China's Eastern QTP (32°49′ N, 102°44′ E, 3450 m a.s.l.). The area has a cold/temperate monsoon climate typical of the continental plateau, with an average temperature of 2.4 °C and average annual rainfall exceeding 700 mm. Each plot was hand-weeded or sprayed with herbicide to suppress the continuous development of grass and weeds. To prepare it for sowing, the land was ploughed three times using a disk plow, with a working depth of 0.25 m and a width of 0.9 m. In addition, field leveling, trench digging, and plot designing were implemented manually. The soil type of the test

site is subalpine meadow soil, and the detailed nutrient parameters revealed a 0–60 cm soil layer (Table S1).

## 2.2. Experimental Management

The seeds of cultivar *K. melanthera* Aba were donated by the Sichuan Academy of Grassland Sciences, Sichuan province of China. This cultivar was bred by mass selection on native wild germplasm of the Northwest Sichuan Plateau located east of the QTP. Moreover, this grass, a tufted plant, has an inflorescence of spike shapes with a large number of villi on its surface (Figure 1). They had a germination percentage of about 86% and a thousand-grain weight of 4.95 g. In June 2014, the seeds were manually harvested and stored at −20 °C. During the first year of the trial (2015), most *K. melanthera* plants were prevented from heading and flowering, and could only grow vegetatively, which is common for most perennial native grasses in the QTP. As a result, no data on seed production were gathered in 2015, and data from 2016 and 2017 were used instead.

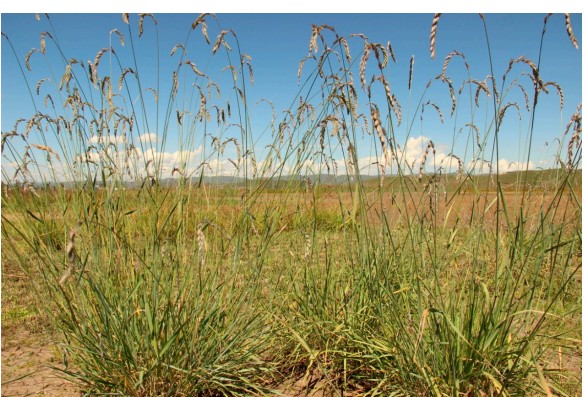

**Figure 1.** *Kengyilia melanthera*. cv. Aba.

## 2.3. Experimental Design

A total of 32 experimental plots were established in this experiment using a randomized complete block design for eight seed growing dates that were regarded as harvest time points. In addition, the seeds were harvested every 72 h from the 15th to 36th day after peak anthesis: $D_1$—15th day after peak anthesis, $D_2$—18th day after peak anthesis, $D_3$—21st day after peak anthesis, $D_4$—24th day after peak anthesis, $D_5$—27th day after peak anthesis, $D_6$—30th day after peak anthesis, $D_7$—33rd day after peak anthesis, $D_8$—36th day after peak anthesis. The date of peak anthesis was determined when 50% of the plants reached the flowering stage. At each harvest time point, four replication plots were included. On 20 May 2015, the streak seeding technique was used to complete the sowing, with a seed sowing density of 20.0 kg·hm$^{-2}$ and a depth of 2–3 cm, respectively. Each plot had a sowing area of 20 m$^2$ (5 m × 4 m), with eight rows of identical distances separated by 50 cm. Furthermore, no sowing was carried out within 1 m of the plot's border to protect against the marginal effects.

## 2.4. Evaluations

### 2.4.1. Seed Water Content, Fresh Weight, and Dry Weight

At different harvest times, seeds from the respective plots were harvested separately to form a seedbank. To determine seed fresh weight (FW), dry weight (DW), and water content (WC), 100 seeds were randomly selected from the seedbank. The fresh weights were calculated using a weighing scale with a sensitivity of 1/10,000. The dry weight was obtained by drying fresh seeds at 65 °C to a constant weight, followed by oven drying at 105 °C for 0.5 h. The seed WC of each treatment was estimated for four replications using the FW and DW values of the seeds, and the calculation formula is as follows [22]:

$$WC\ (\%) = (FW\ (g) - DW\ (g))/FW\ (g) \times 100\%  \tag{1}$$

### 2.4.2. Thousand-Grain Weight

Thousand-grain weight (TGW) was calculated for the corresponding plots of the $D_1$–$D_8$ period by randomly selecting 1000 substantially complete seeds to be sun-dried to a constant weight and subsequently weighed [23].

### 2.4.3. Seed Yield

For the matched plots with varied harvest times, a one-meter-long row segment was chosen at random and all its seeds were harvested. However, the boundary of the plot was not used to test samples. The seed yield (SY) unit area was calculated by threshing, washing, and naturally sun drying these seeds to a WC of around 150 g·kg$^{-1}$ and then correctly weighing them [23].

### 2.4.4. Seed Germination Potential and Generation Percentage

After harvesting, the seeds need to be stored at $-20$ °C for at least 60 days to lessen or prevent dormancy in newly harvested seeds. Before the experiment, the seeds were refrigerated at 5 °C for 7 days and then placed at room temperature for 12 h. Thereafter, the seeds were placed into a petri dish with two layers of filter paper and distilled water, which was added in moderation to just submerge the seeds. Following this, the petri dishes were placed in a plant growth chamber with an average temperature of 24/18 °C day/night and a photoperiod of 14/10 h day/night. Four repetitions of fifty seeds per harvest time were tested for germination potential (GP) at the 7th day and generation percentage (GEP) at the 12th day [24].

### 2.4.5. Seed Accelerated Aging Germination Percentage

Four repetitions of 250 complete and whole seeds per harvest time were wrapped with gauze and placed in the inner side of an aging tank. The temperature of the outer tank was kept at 40 °C and treated for 72 h to yield the seed accelerated aging germinating percentage (AAGP) [25].

### 2.4.6. Dehydrogenase Activity and Acid Phosphoesterase Activity

Triphenyl tetrazolium chloride was used to determine dehydrogenase activity (DA) [26]. The sodium p-nitrophenol phosphate technique was used to assess acid phosphoesterase activity (APA) [27]. For these two indices, four repetitions were performed.

### 2.5. Statistics Analysis

One-way and two-way analyses of variance was performed using the IBM SPSS Statistics 20.0.0 method (N.H. Nie, www.ibm.com/software/analytics/SPSS accessed on 29 October 2022). A general linear model with a $p < 0.05$ significance level was applied to a two-way ANOVA study to analyze the combined influence of eight harvest times and trial years on seed quality. After multiple hypothesis testing at the $p < 0.05$ level of significance, Bonferroni correction was used. Heatmaps and other figures were created using TBtools (X. Rui, https://github.com/CJ-Chen/TBtools accessed on 29 October 2022) and GraphPad Prism 8.0.2 (H. Moltusky, www.graphpad.com accessed on 29 October 2022) application.

The membership function approach was utilized to comprehensively evaluate the optimal harvest time point of *K. melanthera*. The computation of membership function can be broken into forward and backward, $y = (x_i - x_{min})/(x_{max} - x_{min})$, and $y = 1 - (x_i - x_{min})/(x_{max} - x_{min})$, respectively, where the $x_i$ represents the value of a certain index, while $x_{max}$ and $x_{min}$ define its peak and bottom value based on the identical index of seed traits.

## 3. Results

### 3.1. Effects of Harvest Time on Seed Quality and Yield

From the 15th day after peak anthesis ($D_1$, control), seeds were collected every 3 days to record the growth process and determine whether harvest time affects seed yield (WC, TGW, SY) and quality (SGP, GP, AAGP, DA, and APA) of *K. melanthera*. The results show that

prolonging the harvest time had a significant effect on the seed yield and quality indices in both trial years ($p < 0.05$, Table 1). Nonetheless, there was no significant difference ($p > 0.05$) for the same indices of the agronomic properties of the same harvest time across the trial years.

**Table 1.** Analysis of variance of the influence of different days after peak anthesis on harvested seed yield and quality indices of *K. melanthera*.

| Index | Source of Variance | Df | F-Ratio | *p*-Value |
|---|---|---|---|---|
| WC (%) | Year (Y) | 1 | 0.074 | 0.787 |
| | Days after peak anthesis (D) | 7 | 658.684 | 0.000 |
| | Interaction Y × D | 7 | 0.505 | 0.826 |
| SY (kg·hm$^{-2}$) | Year (Y) | 1 | 0.0380 | 0.845 |
| | Days after peak anthesis (D) | 7 | 2315.693 | 0.000 |
| | Interaction Y × D | 7 | 0.173 | 0.990 |
| TGW (g) | Year (Y) | 1 | 0.075 | 0.785 |
| | Days after peak anthesis (D) | 7 | 1646.262 | 0.000 |
| | Interaction Y × D | 7 | 0.795 | 0.595 |
| GEP (%) | Year (Y) | 1 | 0.004 | 0.952 |
| | Days after peak anthesis (D) | 7 | 73.023 | 0.000 |
| | Interaction Y × D | 7 | 0.148 | 0.994 |
| GP (%) | Year (Y) | 1 | 0.065 | 0.800 |
| | Days after peak anthesis (D) | 7 | 63.310 | 0.000 |
| | Interaction Y × D | 7 | 0.293 | 0.953 |
| AAGP (%) | Year (Y) | 1 | 0.064 | 0.832 |
| | Days after peak anthesis (D) | 7 | 70.345 | 0.000 |
| | Interaction Y × D | 7 | 0.253 | 0.729 |
| DA (µg·mL$^{-1}$) | Year (Y) | 1 | 0.055 | 0.815 |
| | Days after peak anthesis (D) | 7 | 131.304 | 0.000 |
| | Interaction Y × D | 7 | 0.588 | 0.369 |
| APA (nmol·min$^{-1}$. 50 seeds) | Year (Y) | 1 | 0.027 | 0.870 |
| | Days after peak anthesis (D) | 7 | 355.659 | 0.000 |
| | Interaction Y × D | 7 | 0.688 | 0.681 |

$p < 0.05$ is considered as a significant difference between different days after peak anthesis and year based on Bonferroni's *t*-test. WC, water content; SY, seed yield; TGW, thousand-grain weight; SGP, germination percentage; GP, germination potential; AAGP, accelerated aging germination percentage; DA, dehydrogenase activity; APA, acid phosphoesterase activity; Df, degrees of freedom.

### 3.2. Effect of Harvest Time on Seed Water Content, Fresh Weight, and Dry Seed Weight

The water content (WC), fresh weight (FW), and dry weight (DW) were used to describe the yield level of seeds from different *K. melanthera* harvest times. As the harvest times prolonged, the seed WC in different years exhibited a decreasing trend from the 15th to the 36th day after peak anthesis (D1–D8) period. The lowest values were observed under 30th day after peak anthesis (D6) period: 36.01% in 2016 and 36.04% in 2017 (Figure 2a). However, the fluctuation tendency of FW and DW were opposite to the WC, and reached their peak value at D6 with 0.815 g, 0.518 g in 2016 and 0.805 g, 0.515 g in 2017, respectively (Figure 2b). Additionally, the WC, FW, and DW at D1 were significantly different compared with these values between D2–D8 ($p < 0.05$), while there was minimal change between D6 and D8 ($p > 0.05$). The ideal harvest time for the three seed yield-related traits thus appeared to be the 30th day after peak anthesis.

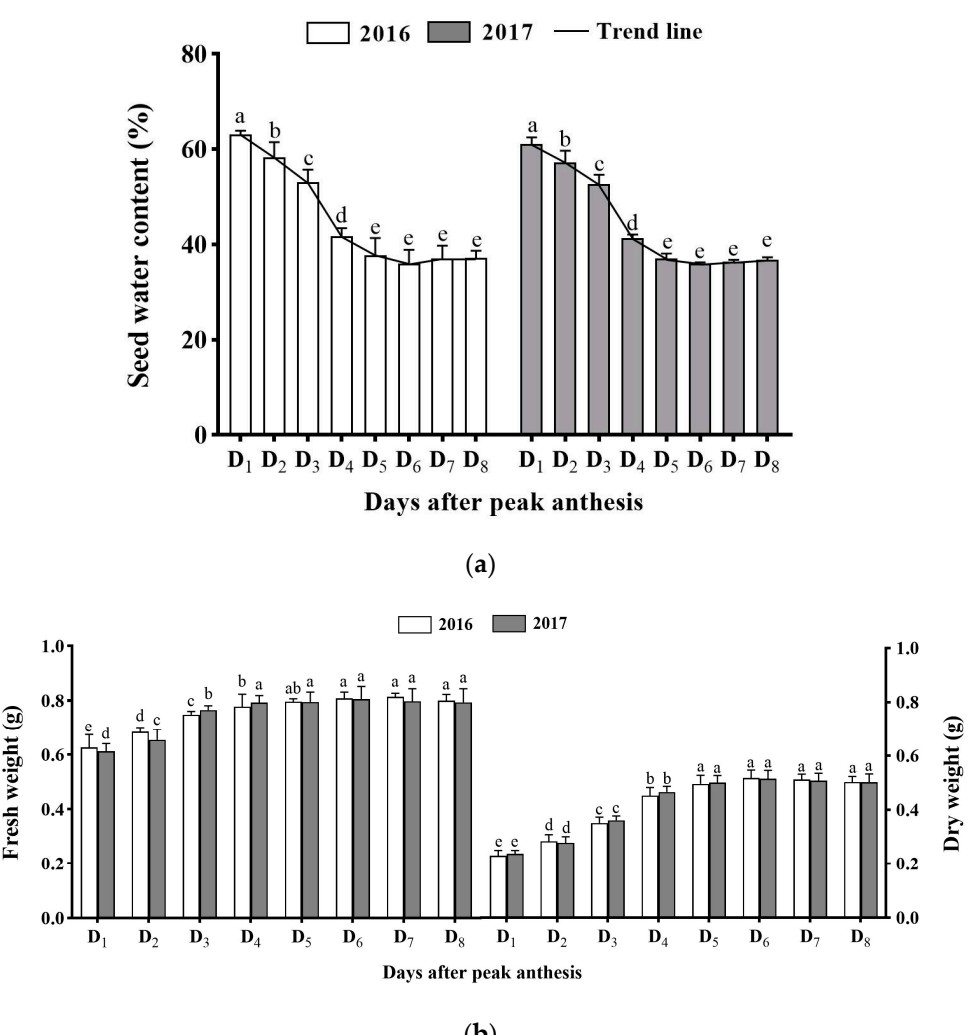

(**a**)

(**b**)

**Figure 2.** The seed water content (**a**) and fresh and dry weights of seeds (**b**) (mean ± standard error, $n$ = 4) of *K. melanthera* as influenced by days after peak anthesis of seeds during the trial years of 2016–2017. For each day after peak anthesis, values with the different letter are significantly different based on Bonferroni's *t*-test ($p < 0.05$). The standard error of mean is shown by a vertical bar. Note: $D_1$—15th day after peak anthesis (control), $D_2$—18th day after peak anthesis, $D_3$—21st day after peak anthesis, $D_4$—24th day after peak anthesis, $D_5$—27th day after peak anthesis, $D_6$—30th day after peak anthesis, $D_7$—33rd day after peak anthesis, $D_8$—36th day after peak anthesis.

*3.3. Effect of Harvest Time on Seed Yield and Thousand-Grain Weight*

During the trial years, the seed yield (SY) (kg·hm$^{-2}$) and thousand-grain weight (TGW) (g) of *K. melanthera* exhibited a similar trend as maturity proceeded (Figure 3a,b). During $D_1$–$D_6$, the SY increased steadily, from 802.44 kg·hm$^{-2}$ to 1773.67 kg·hm$^{-2}$ in 2016 and 842.44 kg·hm$^{-2}$ to 1847.42 kg·hm$^{-2}$ in 2017 (Figure 3a). Moreover, TGW of seven harvest periods ($D_2$ to $D_8$) differed significantly from $D_1$ (Figure 3b, $p < 0.05$). Notably, the TGW in $D_6$, with a maximum of eight harvest times, increased by 143.81% in 2016 and 133.96% in 2017 when compared with $D_1$ ($p < 0.05$). Therefore, the 30th day after peak anthesis was considered as the suitable period for developing a comparatively high level of SY (kg·hm$^{-2}$) and TGW (g) in *K. melanthera*.

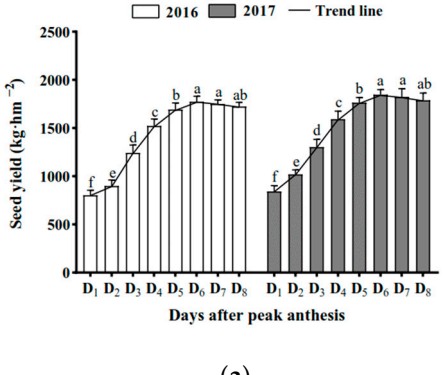
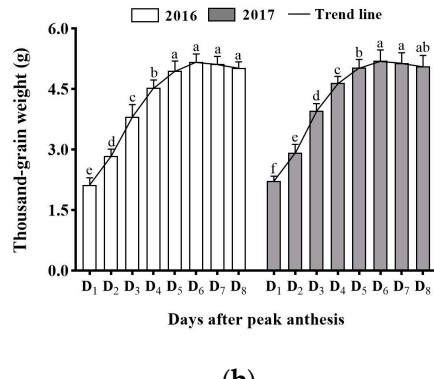

(**a**)                                                           (**b**)

**Figure 3.** The seed yield (**a**) and thousand-grain weight (**b**) (mean $\pm$ standard error, *n* = 4) of *K. melanthera* as influenced by days after peak anthesis of seeds during the trial years of 2016–2017. For different days after peak anthesis, values with the different letter are significantly different based on Bonferroni's *t*-test (*p* < 0.05). The standard error of the mean is shown by a vertical bar. Note: $D_1$—15th day after peak anthesis (control), $D_2$—18th day after peak anthesis, $D_3$—21st day after peak anthesis, $D_4$—24th day after peak anthesis, $D_5$—27th day after peak anthesis, $D_6$—30th day after peak anthesis, $D_7$—33rd day after peak anthesis, $D_8$—36th day after peak anthesis.

*3.4. Effect of Harvest Time on Seed Quality*

It was critical to evaluate seed quality based on the germination percentage (GEP) and germination potential (GP) of seeds. Higher seed quality indicates greater seed vigor and yield potential. In 2016 and 2017, the GEP and GP of seeds harvested before the 24th day after peak anthesis ($D_4$) were less than 80%. This was due to the slightly shorter growth time and a deficiency in dry matter. However, they showed a tendency of rising first and then declining, and their value peaked at $D_6$ (Table 2). During the two-year experiment period, the accelerated aging germination percentage (AAGP) of seeds maintained a progressively growing trend (Table 2). Under normal conditions, seed viability and enzyme activity gradually increased with growing time. The dehydrogenase activity (DA) of seeds significantly increased on the 18th day after peak anthesis ($D_2$), reaching a maximum of 16.23 µg·mL$^{-1}$ and 16.52 µg·mL$^{-1}$ at $D_6$ of 2016 and 2017, respectively (Table 2). The variation pattern of the acid phosphoesterase activity (APA) was similar to that of GEP and GP, reaching a maximum at $D_6$ in the two trial years (Table 2). Furthermore, the APA of seeds at each harvest time increased significantly in 2017 compared with the previous year. It is worth mentioning that the four seed quality indices (GEP, GP, DA, and APA) for seeds harvested on the 33rd day after the peak anthesis ($D_7$) period begin to decline at varying degrees. However, their values were higher than those of $D_1$. Therefore, the optimal harvest time was the 30th to 33rd day following peak anthesis to guarantee that seed yield and quality indices were generally at their highest.

**Table 2.** Seed quality of *K. melanthera* as influenced by different days after peak anthesis of seeds during the 2016–2017 trial years.

| Year | Days after Peak Anthesis | GEP (%) | GP (%) | AAGP (%) | DA (µg·mL$^{-1}$) | APA (nmol·min$^{-1}$. 50 Seeds) |
|------|--------------------------|---------|--------|----------|-------------------|----------------------------------|
| 2016 | $D_1$ | 69.50 e | 67.00 e | 21.50 f | 9.49 e | 1.24 e |
|      | $D_2$ | 72.00 d | 68.00 e | 34.50 e | 9.59 e | 1.27 e |
|      | $D_3$ | 77.50 c | 74.50 d | 45.00 d | 11.51 d | 1.56 d |
|      | $D_4$ | 79.00 c | 76.00 d | 53.50 c | 13.46 c | 1.74 cd |
|      | $D_5$ | 83.00 b | 81.50 bc | 58.50 b | 14.66 b | 1.87 c |
|      | $D_6$ | 87.00 a | 85.00 a | 66.00 a | 16.23 a | 2.50 a |
|      | $D_7$ | 86.00 a | 83.00 b | 67.00 a | 15.83 a | 2.43 a |
|      | $D_8$ | 82.00 b | 80.50 c | 65.50 a | 15.73 ab | 2.21 b |

**Table 2.** *Cont.*

| Year | Days after Peak Anthesis | GEP (%) | GP (%) | AAGP (%) | DA ($\mu g \cdot mL^{-1}$) | APA ($nmol \cdot min^{-1}$. 50 Seeds) |
|---|---|---|---|---|---|---|
| 2017 | $D_1$ | 70.00 f | 65.50 f | 20.00 f | 9.40 f | 1.36 d |
| | $D_2$ | 71.50 e | 67.50 e | 33.50 e | 9.72 f | 1.51 d |
| | $D_3$ | 77.50 d | 74.00 d | 45.50 d | 11.47 e | 1.62 cd |
| | $D_4$ | 80.00 c | 77.50 c | 55.50 c | 12.44 d | 1.82 c |
| | $D_5$ | 83.50 ab | 80.50 b | 62.50 b | 13.97 c | 2.18 b |
| | $D_6$ | 86.50 a | 83.50 a | 65.00 a | 16.52 a | 2.55 a |
| | $D_7$ | 85.00 a | 83.00 a | 64.00 ab | 16.19 a | 2.46 a |
| | $D_8$ | 82.00 b | 80.00 b | 65.50 a | 15.60 b | 2.27 b |

The result of the determination is indicated by mean ± standard error (*n* = 4). In the same column, mean ± standard error followed by different lower case letters (a, b, c, d, e, f) means that there is a significant difference in the values of different days after peak anthesis (Bonferroni, $p < 0.05$). GEP, germination percentage; GP, germination potential; AAGP, accelerated aging germination percentage; DA, dehydrogenase activity; APA, acid phosphoesterase activity; $D_1$—15th day after peak anthesis (control), $D_2$—18th day after peak anthesis, $D_3$—21st day after peak anthesis, $D_4$—24th day after peak anthesis, $D_5$—27th day after peak anthesis, $D_6$—30th day after peak anthesis, $D_7$—33rd day after peak anthesis, $D_8$—36th day after peak anthesis.

### 3.5. Analysis of Membership Function and Heatmap

The variation in the seed yield and quality characteristic indices of *K. melanthera* revealed some level of correlation based on different days after peak anthesis, indicating that a single measure cannot entirely explain the influence of harvest time on plant reproduction and seed production. A combined study of several indicators was, therefore, undertaken to make the identification of optimal harvest time more reliable and precise. The membership function values were calculated based on the values of each seed characteristic indices, which were measured on different days after peak anthesis, including five seed yield indices (MC, FW, DW, TGW, SY) and five seed quality indices (GEP, GP, AAGP, DA, APA, Tables 3 and S2). Higher membership function values demonstrate a more noticeable effect on seed quality and quantity of *K. melanthera*. Eight harvest times were sorted during the trial years of 2016–2017, with $D_6$ being the highest rated, followed closely by $D_7$ (Tables 3 and S2). Therefore, the ideal harvest time in both trial years is proposed to be the 30th day after peak anthesis ($D_6$, followed closely by $D_7$).

**Table 3.** The influences of multi-indicator analysis of membership functions on *K. melanthera* on different days after peak anthesis during the 2016 trial year.

| Parameter | Days after Peak Anthesis | | | | | | | |
|---|---|---|---|---|---|---|---|---|
| | $D_1$ | $D_2$ | $D_3$ | $D_4$ | $D_5$ | $D_6$ | $D_7$ | $D_8$ |
| WC (%) | 0.00 | 0.17 | 0.37 | 0.79 | 0.94 | 1.00 | 0.96 | 0.96 |
| FW (g) | 0.00 | 0.31 | 0.65 | 0.80 | 0.90 | 0.97 | 1.00 | 0.92 |
| DW (g) | 0.00 | 0.19 | 0.42 | 0.77 | 0.92 | 1.00 | 0.98 | 0.95 |
| SY ($kg \cdot hm^{-2}$) | 0.00 | 0.10 | 0.45 | 0.74 | 0.92 | 1.00 | 0.98 | 0.95 |
| TGW (g) | 0.00 | 0.24 | 0.55 | 0.79 | 0.93 | 1.00 | 0.98 | 0.95 |
| GEP (%) | 0.00 | 0.14 | 0.46 | 0.54 | 0.77 | 1.00 | 0.94 | 0.71 |
| GP (%) | 0.00 | 0.06 | 0.42 | 0.50 | 0.78 | 1.00 | 0.89 | 0.75 |
| AAGP (%) | 0.00 | 0.29 | 0.52 | 0.70 | 0.81 | 0.98 | 1.00 | 0.97 |
| DA ($\mu g \cdot mL^{-1}$) | 0.00 | 0.01 | 0.30 | 0.59 | 0.77 | 1.00 | 0.94 | 0.93 |
| APA ($nmol \cdot min^{-1}$. 50 seeds) | 0.00 | 0.02 | 0.25 | 0.40 | 0.50 | 1.00 | 0.94 | 0.77 |
| Rank | 8 | 7 | 6 | 5 | 4 | 1 | 2 | 3 |

WC, water content; FW, fresh weight; DW, dry weight; SY, seed yield; TGW, thousand-grain weight; GEP, germination percentage; GP, germination potential; AAGP, accelerated aging germination percentage; DA, dehydrogenase activity; APA, acid phosphoesterase activity; $D_1$—15th day after peak anthesis (control), $D_2$—18th day after peak anthesis, $D_3$—21st day after peak anthesis, $D_4$—24th day after peak anthesis, $D_5$—27th day after peak anthesis, $D_6$—30th day after peak anthesis, $D_7$—33rd day after peak anthesis, $D_8$—36th day after peak anthesis.

The heatmap and hierarchical cluster analysis were used to elaborate the up-regulated or down-regulated indices for seed quality or yield. Up-regulation of both $D_6$ and $D_7$ was most noticeable in 2016, followed by $D_8$, and $D_5$ (Figure 4a). Consequently, $D_6$ and $D_7$ were classified together based on hierarchical cluster analysis. In both trial years, the changing pattern of each index and harvest time were similar, and the majority of indices with reasonably considerable up-regulation were assigned to $D_6$ and $D_7$ (Figure 4a,b). Furthermore, the results of hierarchical cluster analysis were similar in both trial years. It is worth noting that FW (g) and TGW (g) showed a significant up-regulation at $D_3$, while WC (%) revealed an opposite expression level compared to other indicators. The heatmaps and extensive analysis of membership function confirmed similar findings based on these two trial years, indicating that the ideal harvest time was the 30th to 33rd day after peak anthesis ($D_6$–$D_7$).

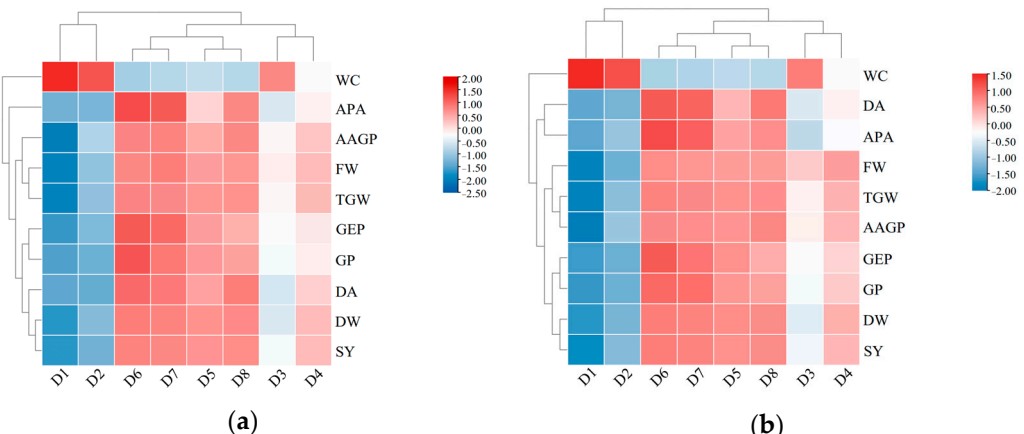

(**a**)                    (**b**)

**Figure 4.** Heatmaps showing a $\log_2$-fold change in seed yield and quality observed in *K. melanthera* based on different days after peak anthesis during the trial years 2016 (**a**) and 2017 (**b**). Up-regulated and down-regulated indices are indicated by red and blue, respectively. Note: WC, water content; FW, fresh weight; DW, dry weight; SY, seed yield; TGW, thousand-grain weight; GEP, germination percentage; GP, germination potential; AAGP, accelerated aging germination percentage; DA, dehydrogenase activity; APA, acid phosphoesterase activity; $D_1$—15th day after peak anthesis (control), $D_2$—18th day after peak anthesis, $D_3$—21st day after peak anthesis, $D_4$—24th day after peak anthesis, $D_5$—27th day after peak anthesis, $D_6$—30th day after peak anthesis, $D_7$—33rd day after peak anthesis, $D_8$—36th day after peak anthesis.

## 4. Discussion

Harvest timing is critical in agricultural production, and is often a deciding factor in yield and grass seed quality [28]. Poor harvest timing might be one of the primary reasons for low yield and seed quality. The seeds harvested too early may have poorer maturity and vigor value, whereas seeds harvested too late may suffer from grain loss and deterioration [29]. A large number of grass species have some degree of seed fall and concentrate mainly on the late stage of maturity. As a result, determining the best time to harvest is crucial for reducing seed loss and improving seed yield and quality.

Higher water content (WC) prevents the seeds from falling off, while lower WC hinders the various physiological indices of the seeds from achieving the ideal state [30]. In addition, WC was negatively correlated with seed shattering but positively correlated with the accumulation of the dry matter of the endosperm [31,32]. In other words, the higher the seed maturity, the lower the WC, the drier the content matter, and the greater the grain shattering degree [33]. According to the findings of a significant number of herbage seed harvest time studies, WC is considered an appropriate index for determining the optimal harvest time and effective storage [34]. We discovered that continuous observation of the change in seed WC can offer a basis for timely seed harvesting. For most forage grasses, seed harvesting can take place when the WC is lowered to about 40%, although

this varies depending on the species [35]. The outcome of the present study is similar to prior grass study results. For instance, ryegrass and fescue species can be harvested when seeds WC of the seed is around 40%. When the water content was 40%, tall fescue seeds had the highest yield and maximum germination rate, and their natural shedding was the lowest [36]. When the seed was harvested, the WC of *Paspalum* seeds is approximately 35 to 45%. When harvested at the appropriate time, the WC of *Elymus* seeds is between 40 to 35%. At the appropriate harvest period, the WC of *Agropyron* seeds is between 30 to 20%. Early harvesting of seeds will result in poor storage and limited life power [37]. On the other hand, late-harvested seeds have greater dormancy and higher viability [38]. This study found that the WC of *K. melanthera* seeds decreased steadily with maturity, reaching the lowest value at $D_6$ of less than 37% during trial years.

Another recommended way for determining the best harvest time is to evaluate seed yield (SY) and thousand-grain weight (TGW) [39]. However, maximum yield and TGW vary greatly depending on the species. On the 38th day after peak anthesis, *Brachiaria decumbens* seeds exhibited the highest SY, while TGW and seed vigor were correspondingly high [40]. The SY, TGW, and germination percentage (GEP) of *Leymus chinensis* seeds grew and then decreased with the extension of harvest time, reaching the peak on the 33rd day after peak anthesis [41]. This is an indication that the optimal harvest time is 33rd days after peak anthesis.

According to the current study, SY and TGW of *K. melanthera* seeds increased gradually with the seed maturation, both reaching the peak values of over 1700 kg·hm$^{-2}$ and 4.5 g, respectively, 30 days after peak anthesis. Although seeds following harvest, such as at $D_6$, showed a minor gain in seed quality and vigor while continuing to prolong harvest time ($D_7$ to $D_8$), the loss of falling grain is significant, therefore early harvest should be preferred. It is worth noting that studies on wild *E. nutans* in Tibet also reached similar conclusions [17].

Seed quality is usually defined by the germination percentage (GEP), germination potential (GP), and accelerated aging germination percentage (AAGP). In general, as these values improve, seed vigor will also increase [42]. Interestingly, when AAGP is compared with GEP, resistance to adversity can be better represented by the experiment environment of AAGR, environments that offer a more reliable recommendation during seed development under certain conditions such as the QTP [43]. As GEP was developed under acceptable climatic conditions, real germination rates in the field or survival are rarely guaranteed [44]. As a result, using AAGP to analyze the potential vitality of seeds is crucial. Grass seed vigor and usefulness are often determined using tests based on dehydrogenase activity (DA) and acid phosphoesterase activity (APA). This is because these two indicators are more useful in revealing the quality and stress resistance of seeds throughout different phases of growth compared with seed germination counts [45]. During the trial years, the GEP, GP, AAGP, DA, and APA of seeds all demonstrated a sustained upward trend with seed maturation, and achieved the optimal value at the 30th day after peak anthesis ($D_6$). After this period ($\sim D_7$–$D_8$), the seed quality and vigor appeared to diminish to varying degrees, which might be due to a combination of external variables and the species' own declining grain. To ensure good seed quality and production, real-time monitoring of falling grain, seed maturity, and meteorological conditions is required.

## 5. Conclusions

In East QTP, *K. melanthera* is considered a valuable perennial grass with several applications. It can withstand cold, drought, and wind erosion. It is commonly used to regenerate grassland vegetation, prevent desertification, and provide feed for livestock. Nonetheless, its development and promotion process has been slowed in the QTP due to its restricted supply and relatively low seed quality. Therefore, a field experiment was carried out for two consecutive years (2016 and 2017) to examine the impacts of seed output and quality of *K. melanthera* based on eight harvest timings. The findings reveal that the 30th day after peak anthesis was the ideal period for harvesting for both trial years. These data uncover a

variety of high-efficiency and viable technical recommendations for improving the seed quality and yield of *K. melanthera* on the Northwest Sichuan plateau.

**Supplementary Materials:** The following supporting information can be downloaded at https://www.mdpi.com/article/10.3390/agronomy13010055/s1. Table S1: Basic nutrient of experimental soil. Table S2: The influences of multi-indicator analysis of membership functions on *K. melanthera* on different days after peak anthesis during the 2017 trial year.

**Author Contributions:** Conceptualization, Y.L. and S.Y.; methodology, X.M.; software, Y.X.; validation, Y.L., S.Y. and S.C.; formal analysis, J.F.; investigation, J.Z.; resources, Y.L. and X.M.; data curation, C.Z.; writing—original draft preparation, S.Y. and Y.X.; writing—review and editing, Y.L. and S.Y.; visualization, X.L.; supervision, M.Y.; project administration, S.B. and X.M.; funding acquisition, Y.L. All authors have read and agreed to the published version of the manuscript.

**Funding:** This research was funded by the Sichuan Provincial Key Science and Technology Project (2022YFQ0076, 2023SYSX0014 and 2019YFN0170).

**Data Availability Statement:** Data are contained within the article or Supplementary Materials.

**Acknowledgments:** We thank the laboratory staff in the College of Grassland Science and Technology, Sichuan Agricultural University, Chengdu, China.

**Conflicts of Interest:** The authors declare no conflict of interest.

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
