# Peer review of "Effect of Harvest Time on the Seed Yield and Quality of Kengyilia melanthera"

_agronomy, doi:10.3390/agronomy13010055_

Round 1

Reviewer 1 Report

This manuscript describes an experiment to investigate the maturity of seed of Kengyilia following anthesis on the germination and quality of the seed.  Kengyilia is a useful species for renovating degraded land, so producing adequate supplies of high viability seed is critical.  Thus, the results of this study represent important information for land renovation.  I would recommend publication of this manuscript in Agronomy, but considerable revision is required as described in the following. 

Some general editing needs to be done to improve the English grammar.  For example, all verb tenses should be in the past, not the present or future, as the research has already been conducted. 

In the Materials and Methods section of the manuscript, evaluations of a number of seed germination, yield and quality parameters are listed.  However, there is no explanation of why these evaluations are being done.  All of the parameters and their importance should be described in the Introduction section of the manuscript to make it easier for the reader of the paper to understand what if being done.  These parameters are mentioned later in the Discussion section of manuscript, but it would make the paper clearer if they were described in the Introduction. 

Section 2.3 Experimental Design – How was peak anthesis determined?  Individual plants of a grass species may reach anthesis over a period of several days, even as long as week in some cases.  So, the authors should describe how they determined the date of peak anthesis. 

Lines 127, 131 and elsewhere – It is not necessary to state ‘At each harvest time, four replications were performed’ after each measurement.  Under the Experimental Design section, it is stated that there were four replications in the experiment, so one assumes that all replications would be sampled and evaluated.

Lines 203-205 – It is stated that seed yield increased steadily from D1 to D8.  In fact it only increased to D6 with no further increase in D7 and D8

Lines 234, 235 – In most cases quality parameters did not significantly decline until D8.  D6 and D7 were seldom different. 

Lines 297 – 300.  It is indicated that water content was negatively associated with seed shattering.  It is unclear whether this is based on other studies or the present study.  Was seed shattering measured in the present study?  If so, data should be presented; if not, it would have been very useful to have this information as this is the key factor in being able to harvest a large quantity of seed.  If seed shattering is an issue for Kengyilia, then perhaps D6 would not be ideal stage to harvest seed.  D4 and D5 harvested seed had germination and quality almost as good as D6, so harvesting earlier could avoid shattering losses and still result in high quality seed.  This should be discussed in the manuscript. 

Line 306 – It is stated that in most cases 40% is the optimum seed moisture.  In the present study D6 was 37% moisture.  D5 appears to be closer to 40%, so for this parameter would be the best stage. 

Author Response

Response to Reviewer 1 Comments

Point 1: Some general editing needs to be done to improve the English grammar.  For example, all verb tenses should be in the past, not the present or future, as the research has already been conducted.

Response 1: Thanks for the reviewer’s question. We have improved the sentences with incorrect English grammar in line 30, 62, 63, and 92-94

Point 2: In the Materials and Methods section of the manuscript, evaluations of a number of seed germination, yield and quality parameters are listed.  However, there is no explanation of why these evaluations are being done.  All of the parameters and their importance should be described in the Introduction section of the manuscript to make it easier for the reader of the paper to understand what if being done.  These parameters are mentioned later in the Discussion section of manuscript, but it would make the paper clearer if they were described in the Introduction.

Response 2: Thanks for the reviewer’s suggestion. We have added descriptions of all parameters in the Introduction section, see line 82-87 for more details.

Point 3: Section 2.3 Experimental Design – How was peak anthesis determined?  Individual plants of a grass species may reach anthesis over a period of several days, even as long as week in some cases.  So, the authors should describe how they determined the date of peak anthesis.

Response 3: Thanks for the reviewer’s question. The date of peak anthesis was determined when 50% of the plants reached the flowering stage. In addition, we added the description of how to determine the date of peak anthesis in line 136-137.

Point 4: Lines 127, 131 and elsewhere – It is not necessary to state ‘At each harvest time, four replications were performed’ after each measurement.  Under the Experimental Design section, it is stated that there were four replications in the experiment, so one assumes that all replications would be sampled and evaluated.

Response 4: Thanks for the reviewer’s suggestion. We have deleted unnecessary statements.

Point 5: Lines 203-205 – It is stated that seed yield increased steadily from D1 to D8.  In fact it only increased to D6 with no further increase in D7 and D8.

Response 5: Thanks for the reviewer’s suggestion. We have changed “D8” to “D6” in line 241.

Point 6: Lines 234, 235 – In most cases quality parameters did not significantly decline until D8.  D6 and D7 were seldom different.

Response 6: Thanks for the reviewer’s question. According to the reviewer’s suggestion, we have changed “30th” to “30th to 33rd” in line 275.

Point 7: Lines 297 – 300.  It is indicated that water content was negatively associated with seed shattering.  It is unclear whether this is based on other studies or the present study.  Was seed shattering measured in the present study?  If so, data should be presented; if not, it would have been very useful to have this information as this is the key factor in being able to harvest a large quantity of seed.  If seed shattering is an issue for Kengyilia, then perhaps D6 would not be ideal stage to harvest seed.  D4 and D5 harvested seed had germination and quality almost as good as D6, so harvesting earlier could avoid shattering losses and still result in high quality seed.  This should be discussed in the manuscript.

Response 7: Thanks for the reviewer’s question. Seed shattering was not measured in this study. Due this was a clerical error, “In this study” has been changed to “In addition” in line 343. Typically, seed shattering can lead to lower seed yield, while the seed yield in this study reached the peak at D6, rather than D4 or D5. Moreover, although D4 and D5 harvested seed had germination and quality almost as good as D6, the comprehensive evaluation of the membership function revealed that the 30th day after peak anthesis was the optimal harvest time for both 2016 and 2017. Taken together, D6 should still be evaluated as the best harvest time.

Point 8: Line 306 – It is stated that in most cases 40% is the optimum seed moisture.  In the present study D6 was 37% moisture.  D5 appears to be closer to 40%, so for this parameter would be the best stage.

Response 8: Thanks for the reviewer’s question. The optimal water content often varies with species. Based on previous research results, the lower the water content have greater dormancy and higher viability. In this study, the seed water content reached the lowest value at D6. During this time, seed yiled, and thousand-grain weight have also reached the optimal value. Therefore, D6 should be determined as the best stage for this parameter.

Reviewer 2 Report

Dear Authors,

I hope to accept my comment about your manuscript!

Best regards ...

Author Response

Response to Reviewer 2 Comments

Point 1: Line 22: standard germination percentage, germination percentage.

Response 1: Thanks for the reviewer’s suggestion. According to the reviewer’s suggestion, we have changed “standard germination percentage” to “germination percentage” in line 23.

Point 2: Line 29: Keywords: Kengyilia melanthera; Qinghai-Tibet Plateau; harvest time; seed yield; seed quality.: harvest time; Kengyilia melanthera; Qinghai-Tibet Plateau; seed quality ; seed yield;.

Response 2: Thanks for the reviewer’s suggestion. According to the reviewer’s suggestion, we have changed “Keywords: Kengyilia melanthera; Qinghai-Tibet Plateau; harvest time; seed yield; seed quality.” to “harvest time; Kengyilia melanthera; Qinghai-Tibet Plateau; seed quality ; seed yield.” in line 33, and line 34.

Point 3: Line 92-96: it will be present in form Table with table S1.

Response 3: Thanks for the reviewer’s suggestion. According to the reviewer’s suggestion, we have deleted the values that already given in the table S1.

Point 4: Line 98-99: Sichuan Academy of Grassland Science, where (Country)?!.

Response 4: Thanks for the reviewer’s suggestion. According to the reviewer’s suggestion, we have added the country where Sichuan Academy of Grassland Science is located in line 117.

Point 5: Line 108-111: it will be present it: every 72h time from 15 to 36 day……

Response 5: Thanks for the reviewer’s suggestion. We have added “the seeds were harvested every 72 h from 15th to 36th day after peak anthesis” in line 132-133.

Point 6: Line 118-119: You have 100 seeds!! How many replaces used you for fresh weights (FW), dry weight (DW)? 

Point 7: Line 121-122: You have 4 replaces “seed water content (WC)”!! How many seeds used you for seed water content (WC)?

Point 8: you have 100 seed per replicate? Because, in the question (1) you used WC, FW, DW!!!

Response 6-8: Thanks for the reviewer’s suggestion. Indeed, the fresh and dry weights of seeds were determined using the the same batch of seeds. Firstly, the fresh weights were calculated using a weighing scale with a sensitivity of 1/10,000. The seed dry weight was then measured using 65 °C to a constant weight following oven-drying at 105 °C for 0.5 h. Therefore, the determination of fresh weights, dry weight has no correlation with the number of seeds. More information have been described in line 145-148.

Point 9: Line 126: you used sundry.  How long time the sun-dry seeds.

Response 9: Thanks for the reviewer’s question. Stopping when the seeds are sun-dry to a constant weight.

Point 10: Line 135-136: why did you conserved at -20°C which reference or stander you used?

Response 10: Thanks for the reviewer’s question. After harvesting, the seeds were stored at −20 °C to lessen or prevent dormancy in newly harvested seeds. In addition, choi et al.’s  [1] study on Glycine max (L.) Merrill also conserved its seeds at -20°C.

  1. Choi, Y.M.; Yoon, H.; Lee, S.; Ko, H.C.; Desta, K.T. Isoflavones, anthocyanins, phenolic content, and antioxidant activities of black soybeans (Glycine max (L.) Merrill) as affected by seed weight. Rep. 2020, 10, 19960.

Point 11: Line 137 and 143: thereafter, Line 137: 50 seeds were chose Line 143At each harvest time, four replications were per- formed.  Four repetitions of 50 seeds per harvest time were tested for germination potential (GP) at 7d and standard generation percentage (SGP) at 12d.

Response 11: Thanks for the reviewer’s suggestion. We have deleted “50 seeds were chose” and “At each harvest time, four replications were performed”. Moreover, we have changed “The Seeds germinated with germination potential on the 7th day and standard genera-tion percentage on the 12th day following germination.” to “Four repetitions of 50 seeds per harvest time were tested for germination potential at the 7th day and generation percentage at the 12th day” in line 170-175.

Point 12: Line 141: How many of photoperiod of 14/10 h day/night per (μmol m−2 s−1) used you?

Response 12: Thanks for the reviewer’s question. Photoperiod of 14/10 h day/night were set just to the determination of germination potential and generation percentage, cut-off time of this photoperiod regarded as 12th day following sowing.

Point 13: Line 146-149: line 146: For seed aging treatment, 250 complete and whole seeds & Line 149: At each harvest time, four replications were performed.. You can to write: Four repetitions of 250 seeds per harvest time were tested for seed aging etc….

Response 13: Thanks for the reviewer’s suggestion. We have deleted “For seed aging treatment, 250 complete and whole seeds.” and “At each harvest time, four replications were performed”. Furthermore, we have changed “For seed aging treatment, 250 complete and whole seeds etc….” to “Four repetitions of 250 complete and whole seeds per harvest time etc...” in line 177-178.

Point 14: Line 153-154: In each of these indices, four repetitions were performed.  How many was it?

Response 14: Thanks for the reviewer’s question. Respectively here referred to the two indices dehydrogenase activity and acid phosphoesterase activity.

Point 15: Line 156 and Line 160: which version used you for SPSS and GraphPad?

Response 15: Thanks for the reviewer’s suggestion. We have added the version of SPSS and GraphPad in line 189 and line 193.

Point 16: Lina 169: “3.1. The effects of harvest time on seed yield and quality” it is good “3.1. Effects of harvest time on quality seed yield “

Response 16: Thanks for the reviewer’s suggestion. According to the reviewer’s suggestion, we have changed “3.1. The effects of harvest time on seed yield and quality” to “3.1. Effects of harvest time on quality seed yield” in line 201.

Point 17: Line 174 & 175: (P < 0.05, Table 1) line 175 (P > 0.05) ?!

Response 17: Thanks for the reviewer’s question. We have added a more detailed description of P < 0.05 and P > 0.05 in line 212 to 213.

Point 18: Line 179: standard germination percentage ? it is germination percentage (%)

Response 18: Thanks for the reviewer’s suggestion. According to the reviewer’s suggestion, we have changed “standard germination percentage” to “germination percentage”.

Point 19: Table (1S) and Table (2S):& Table (3):

  • WC/% it is WC (%)
  • TGW/g: it is TGW (g)
  • SY/kg.hm-2 it is SY(kg.hm-2)
  • SGP/% it is SGP(%)
  • GP/% it is (GP%)
  • AAGP/% it is AAGP(%)
  • DA/µg.mL-1 it is DA (µg.mL-1)
  • APA/ nmol.mib-1.50seeds it is APA (nmol.mib-1.50seeds)

Response 19: Thanks for the reviewer’s suggestion. According to the reviewer’s suggestion, we have modified these data formats in Table (1), Table (2) & Table (3)

Point 20: Lina 182: “3.2. The effect of harvest time on seed water content, fresh weight, and dry weight”  it is good “3.2. Effects of harvest time on seed water content, fresh and dry seed weight “

Response 20: Thanks for the reviewer’s suggestion. We have changed “3.2. The effect of harvest time on seed water content, fresh weight, and dry weight” to “3.2. Effects of harvest time on seed water content, fresh and dry seed weight” in line 217.

Point 21: Line 188: the fluctuation tendency of FW and DW was were opposite to the WC and reached…

Response 21: Thanks for the reviewer’s suggestion. We have changed “was” to “were” in line 223.

Point 22: Figure 1 A: title of (Y-axis) is “Water content/%”it will be “Seed Water content (%)”and (b) will be in upper not in Bottom. Figure (1A) could be precept in histogram.

Response 22: Thanks for the reviewer’s suggestion. We have changed “Water content/%” to “Seed water content (%)” and moved (b) from bottom to upper and Figure (1a) has been changed to a histogram.

Point 23: Figure 1B: title of (Y-axis) is “ Fresh weight /g (1.0) and Dry weight/g (0.8)” it will be smellier “ Fresh weight (g) and Dry weight(g) and same value of 1.0”  and title for Fresh weight and Dry weight will be in same direction;(b) will be in upper not in Bottom.

Response 23: Thanks for the reviewer’s suggestion. According to the reviewer’s suggestion, we have modified these flaws in Figure 1b.

Point 24: Line 194: “fresh (b) and dry weight (b)”, it will be “fresh and dry weight seeds (b)…

Response 24: Thanks for the reviewer’s suggestion. We have changed “fresh (b) and dry weight (b)” to “fresh and dry weight seeds (b)” in line 229.

Point 25: Line 196-197: Within each days after peak anthesis, values with the different letter are significantly different based on Bonferroni t’s t-test (p < 0.05).

Response 25: Thanks for the reviewer’s suggestion. We have changed “Different lower-case letters on the figure represent the degree of significant differences between different days after peak anthesis (Bonferroni, P < 0.05)” to “Within each days after peak anthesis, values with the different letter are significantly different based on Bonferroni t’s t-test (P < 0.05)” in line 231 to 233.

Point 26: Line 201. “3.3. The effect of harvest time on seed yield and thousand-grain weight” it will be “3.3. Effects of harvest time on seed yield and thousand-grain weight”.

Response 26: Thanks for the reviewer’s suggestion. We have changed “3.3. The effect of harvest time on seed yield and thousand-grain weight” to “3.3. Effects of harvest time on seed yield and thousand-grain weight” in line 238.

Point 27: Line 202: seed yield (SY) (kg.hm-2) and thousand-grain weight (TGW) (g).

Response 27: Thanks for the reviewer’s suggestion. According to the reviewer’s suggestion, we have added “(kg.hm-2)” and “(g)” in line 239 and 240.

Point 28: Figure 2A: title of (Y-axis) is “ Seed yield /kg.hm-2 and TGW/g” it will be smellier “Seed yield (kg.hm-2 ) and TGW(g)” and Figure (2B) could be precept in histogram..

Response 28: Thanks for the reviewer’s suggestion. We have changed “Seed yield /kg.hm-2 and TGW/g” to “Seed yield (kg.hm-2) and TGW(g)” in Figure 2a and Figure (2b) has been changed to a histogram.

Point 29: Line 209 & Line 268: high level of SY (kg.hm-2) and TGW(g) in K. melanthera…

Response 29: Thanks for the reviewer’s suggestion. According to the reviewer’s suggestion, we have modified these flaws in line 246-247, and line 308-309.

Point 30: Line 211-212: different days 211 and 212 after peak anthesis, values with the different letter are significantly different based on Bonferroni t’s t-test (p < 0.05).

Response 30: Thanks for the reviewer’s suggestion. We have changed “Different lower-case letters on the figure represent the degree of significant differences between different days after peak anthesis (Bonferroni, P < 0.05)” to “different days after peak anthesis, values with the different letter are significantly different based on Bonferroni’s t-test (P < 0.05)” in line 250-252.

Point 31: Line 217: “3.4. The effect of harvest on seed quality” it will be “3.4. Effects of harvest on seed quality”.

Response 31: Thanks for the reviewer’s suggestion. We have changed “3.4. The effect of harvest on seed quality” to “3.4. Effects of harvest on seed quality” in line 257.

Point 32: Line 218: standard germination percentage (SGP).

Response 32: Thanks for the reviewer’s suggestion. According to the reviewer’s suggestion, we have changed “standard germination percentage” to “germination percentage” in line 258.

Point 33: Line 202: SGP and.

Response 33: Thanks for the reviewer’s suggestion. According to the reviewer’s suggestion, we have changed all “SGP” to “GEP” in this study.

Point 34: Line 228 : (μg/mL-1 )and 16.52 (μg/mL-1).

Response 34: Thanks for the reviewer’s suggestion. According to the reviewer’s suggestion, we have modified it in line 268.

Point 35: Line 229 and Line 232, Line 241, 255& Table S2, Line 275, Line 283, 321, 331, 334, 336, 343; 358: SGP and.

Response 35: Thanks for the reviewer’s suggestion. According to the reviewer’s suggestion, we have changed all “SGP” to “GEP” in this study.

Point 36: Table 2: SGP (%);  DA (μg/.mL-1); APA  (nmol/  .min. 50 seeds)

Response 36: Thanks for the reviewer’s suggestion. According to the reviewer’s suggestion, We have modified it in Table 2

Point 37: Figure 3 A & B: those figures are small & can you change size ?.

Response 37: Thanks for the reviewer’s suggestion. We have enlarged Figure 3 a & b.

Point 38: Line 281: 2017 (bB).

Response 38: Thanks for the reviewer’s suggestion. According to the reviewer’s suggestion, we have deleted “B” in line 325.

Point 39: Line 376: Sichuan Agricultural University, (where)?! 

Response 39: Thanks for the reviewer’s suggestion. We have added the city and country where Sichuan Agricultural University is located in line 424-426.

Reviewer 3 Report

It is more appropriate to give the results obtained in the abstract with numbers.

In the material part, a visual about the plant must be given.

Line between 48-69…The results of previous studies should be supported by figures and given in the study.

Line 82-83…what is simple, easy-to- implement, and sustainable technical strategies for seed quality and yield? It should not be given implicitly, it should be explained.

LÄ°ne 94-96.. These values are already given in the table, they do not need to be given again in the article.

Line 92… For sowing preparation, was the soil plowed only with a plow or was there no other action? How was the planting done in the area that was plowed only with a plow? What is the plow type, working depth and width? What type of machine was the planting done with? How is the type of sowing?

Line 117… On what basis was the measurement of seed water content, fresh weight, and dry weight made? Source should be included.

Equation 1... Units of unknowns must be given.

When determining the seed yield, if there is a picture taken from the test site, it should be added.

2.4.1 ,2,3,4,5 …..in these titles…. Measurements should be based on sources.

Figure 1 dimensions should be enlarged.

Title 3.4. check it.. Could there be a mistake or omission in the title?

Table 2.. µg mL-1

What is a,b,c,d and e? description should be added to the bottom of the table.

The discussion section was like a literature review. Values found (results) need to be visibly discussed with previous work.

Author Response

Response to Reviewer 3 Comments

Point 1: It is more appropriate to give the results obtained in the abstract with numbers.

Response 1: Thanks for the reviewer’s question. According to the reviewer’s suggestion, we have given the results obtained in the abstract with numbers in line 27 to 30.

Point 2: In the material part, a visual about the plant must be given.

Response 2: Thanks for the reviewer’s question. According to the reviewer’s suggestion, we have added the description about the plant in line 116-121.

Point 3: Line between 48-69…The results of previous studies should be supported by figures and given in the study.

Response 3: Thanks for the reviewer’s question. We have added “At present, only a few research has been targeted to studying the effect of harvest time on seed yield and quality in some sympatric grass species of K. melanthera in QTP.” to line 67 to 69.

Point 4: Line 82-83…what is simple, easy-to- implement, and sustainable technical strategies for seed quality and yield? It should not be given implicitly, it should be explained.

Response 4: Thanks for the reviewer’s suggestion. We have changed “This will help to develop simple, easy-to- implement, and sustainable technical strategies for seed quality and yield of K. melanthera ‘Aba’” to “This will help to determine an optimal harvest time and thus provide a more effective strategy for increasing seed quality and yield of K. melanthera ‘Aba’.” in line 96-98.

Point 5: Line 94-96.. These values are already given in the table, they do not need to be given again in the article.

Response 5: Thanks for the reviewer’s suggestion. According to the reviewer’s suggestion, we have deleted these values that already given in the table.

Point 6: Line 92… For sowing preparation, was the soil plowed only with a plow or was there no other action? How was the planting done in the area that was plowed only with a plow? What is the plow type, working depth and width? What type of machine was the planting done with? How is the type of sowing?

Response 6: Thanks for the reviewer’s question. We used four steps to prepare for sowing. The land was first ploughed, the uneven field was subsequently levelled, and finally digging the trench and plot designing. The land was ploughed three times using disk plow, with a working depth of 0.25 m and a width of 0.9 m. Leveling the field, digging the trench, plot designing, sowing were implemented by manual. In addition, the above information is shown in line 107-109.

Point 7: Line 117… On what basis was the measurement of seed water content, fresh weight, and dry weight made? Source should be included.

Response 7: Thanks for the reviewer’s suggestion. We have added the sourch of seed water content, fresh weight, and dry weight in line 144.

Point 8: Equation 1... Units of unknowns must be given.

Response 8: Thanks for the reviewer’s suggestion. We have added the units of the equation 1.

Point 9: When determining the seed yield, if there is a picture taken from the test site, it should be added.

Response 9: Thanks for the reviewer’s suggestion. We have uploaded three picture taken from the test site in Figure S1. Due to this study was performed 6 years ago, picture of determining the seed yield cannot be provided.

Point 10: 2.4.1 ,2,3,4,5 …..in these titles…. Measurements should be based on sources.

Response 10: Thanks for the reviewer’s suggestion. We have provided the source of these measurements in line 144, 153, 157, 163, and 176.

Point 11: Figure 1 dimensions should be enlarged.

Response 11: Thanks for the reviewer’s suggestion. We have enlarged the dimensions of Figure 1.

Point 12: Title 3.4. check it.. Could there be a mistake or omission in the title?

Response 12: Thanks for the reviewer’s suggestion. We have filled in the omission in title 3.4.

Point 13: Table 2.. µg mL-1

Response 13: Thanks for the reviewer’s suggestion. We have changed “µg/mL” to “µg·mL-1” in Table 2.

Point 14: What is a,b,c,d and e? description should be added to the bottom of the table.

Response 14: Thanks for the reviewer’s question. We have added the description of a,b,c,d and e to the bottom of the table 2.

Point 15: The discussion section was like a literature review. Values found (results) need to be visibly discussed with previous work.

Response 15: Thanks for the reviewer’s suggestion. We have visibly discussed the values found (results) with previous work in line 351-352, and 376-377.

Round 2

Reviewer 3 Report

Line 144. You can not give literatüre in title, it must be in the end of paragraph

Equation 1.. You must give units “i said that my first report “Equation 1... Units of unknowns must be given.” But maybe you did not see it…

“2.4.1 ,2,3,4,5 …..in these titles…. Measurements should be based on sources.” Ä° wrote in first report but author did not add anything.. Since you didn't come up with these measurements, you must add literature. You should include the literature for all measurement methods in the article

The research is a 2-year study. It is really difficult to see the effect in 2 years in studies on seed quality and yield. Therefore, such studies must be at least 3 years of work.
